# Crystallographic character of grain boundaries resistant to hydrogen-assisted fracture in Ni-base alloy 725

John P. Hanson [1], Akbar Bagri[2,3], Jonathan Lind[4,5], Peter Kenesei[6], Robert M. Suter [4], Silvija Gradečak[3] & Michael J. Demkowicz [7]

Hydrogen embrittlement (HE) causes sudden, costly failures of metal components across a wide range of industries. Yet, despite over a century of research, the physical mechanisms of HE are too poorly understood to predict HE-induced failures with confidence. We use non-destructive, synchrotron-based techniques to investigate the relationship between the crystallographic character of grain boundaries and their susceptibility to hydrogen-assisted fracture in a nickel superalloy. Our data lead us to identify a class of grain boundaries with striking resistance to hydrogen-assisted crack propagation: boundaries with low-index planes (BLIPs). BLIPs are boundaries where at least one of the neighboring grains has a low Miller index facet—{001}, {011}, or {111}—along the grain boundary plane. These boundaries deflect propagating cracks, toughening the material and improving its HE resistance. Our finding paves the way to improved predictions of HE based on the density and distribution of BLIPs in metal microstructures.

[1] Department of Nuclear Science and Engineering, Massachusetts Institute of Technology, Cambridge, MA 02139, USA. [2] Department of Civil Engineering, Johns Hopkins University, Baltimore, MD 21218, USA. [3] Department of Materials Science and Engineering, Massachusetts Institute of Technology, Cambridge, MA 02139, USA. [4] Department of Physics, Carnegie Mellon University, Pittsburgh, PA 15213, USA. [5] Engineering Directorate, Lawrence Livermore National Laboratory, Livermore, CA 94550, USA. [6] Advanced Photon Source, Argonne National Laboratory, Argonne, IL 60439, USA. [7] Materials Science and Engineering, Texas A&M University, College Station, TX 77843, USA. Correspondence and requests for materials should be addressed to M.J.D. (email: demkowicz@tamu.edu)

Hydrogen embrittlement (HE) of engineering alloys is characterized by a loss of ductility and sudden, often unexpected fracture[1] causing an ever widening spectrum of component failures[2]. In the energy industry alone, HE has been reported in petrochemical[3,4], nuclear[5,6], renewables[7], and hydrogen storage technologies[8]. Hydrogen (H)-related failures also affect the transportation and construction industries[9,10]. They have been reported in medical and dental devices[11–13]. The severity of HE tends to increase with alloy strength[14], leaving advanced alloys particularly susceptible. For example, Ni-base alloy 725 is designed for high strength and corrosion resistance[15], but is nevertheless susceptible to HE[3]. Improved strategies for predicting and preventing HE require deeper understanding of its physical origins. The present study elucidates the relationship between crystallographic character and HE susceptibility of individual grain boundaries (GBs) in alloy 725 using high-energy synchrotron X-ray diffraction and tomography.

HE of metals has been known and continuously investigated for well over a century[1]. While much is now understood about HE[16–19], a steady stream of new insights into its causes evidences that this understanding is still far from complete[20–22]. In particular, it is now widely appreciated that HE does not arise from just one single physical mechanism[17], but is rather the collective outcome of H interactions with all the defects and microstructural features in a material:[23,24] dislocations[25–27], point defects[28,29], GBs[14,21], matrix–precipitate interfaces[30], and free surfaces[31]. Nevertheless, current understanding of H interactions with individual defects and microstructural features is limited and rarely predictive.

For example, while H is known to reduce the cohesive strength of GBs[31–33] and to ease the nucleation of dislocations[34–36], current knowledge is not sufficient to calculate whether bond breaking or dislocation emission will prevail at the tip of a propagating crack in a H-charged metal, precluding predictions of ductile-to-brittle transition using models such as that of Rice and Thomson[37,38]. Similarly, while it is known that H-assisted fracture is intergranular in many metals[20,39–43], current knowledge is not sufficient to predict which GBs are most or least susceptible to HE, hindering microstructure-informed lifetime predictions. The goal of the present work is to advance understanding of HE by investigating the HE susceptibility of individual GBs in Ni-base alloy 725.

H-assisted fracture in Ni-base alloys is usually intergranular[20,40,41]. Thus, GBs are these alloys' weakest links, the microstructural features that are most susceptible to HE. Understanding their deformation and fracture behavior in the presence of H is key to better lifetime predictions and to the design of HE-resistant microstructures. Several investigations have demonstrated the effect of impurity segregation on intergranular HE in Ni-base alloys[41,44], while a pioneering in situ transmission electron microscopy (TEM) investigation by Robertson et al. highlighted the effect of H on slip transmission and its role in GB fracture[45]. Previous investigations of GB HE, however, do not account for the variability of GB properties—and therefore GB–H interactions—arising from GB crystallographic character.

GB structure depends at minimum on five crystallographic parameters:[46] three to describe the relative misorientation, **R**, of the adjoining grains and two for the GB plane normal vector, **n̂**. A growing body of evidence shows that many GB properties depend on all five of these parameters[47–50]. In the context of GB–H interactions, Lawrence et al. identified differences between H-assisted slip transmission at Σ3 twin boundaries and random boundaries[40]. Bechtle et al. showed that thermo-mechanical processing to increase the number of twin-related GBs markedly reduces the HE susceptibility of polycrystalline Ni[14]. Seita et al.

demonstrated that coherent Σ3 twin boundaries are preferential sites for crack initiation in a H-charged Ni-base alloy, yet are also especially resistant to crack propagation, revealing an unexpectedly nuanced dependence of HE on GB character[21]. However, none of the above-mentioned studies fully account for GB character: they primarily distinguish Σ3 twins from the rest of the GBs in the microstructure.

By contrast, the present work relates HE susceptibility to the full, five-parameter crystallographic character of individual GBs. We use high-energy diffraction microscopy (HEDM) to image grain shapes and orientations in polycrystals non-destructively and in 3D. Concurrent X-ray absorption tomography (XRAT) measurements allow us to connect individual GBs to crack morphology unambiguously. Using this combined approach, we show that GBs whose constituent grains meet along crystallographic facets with low Miller indices—which we term "boundaries with low-index planes" (BLIPs)—are especially resistant to crack propagation in the presence of H. BLIPs include coherent twin boundaries (CTBs) as well as GBs that do not have low Σ values. Thus, our work puts into question the utility of categorizing GBs as special or general based on Σ numbers alone[51] and emphasizes the importance of considering the full GB character—including the GB plane[52]—in accounting for GB properties. These results motivate the development of new failure prediction and microstructure design techniques that recognize the resistance of BLIPs to H-assisted fracture.

## Results

**Crack deflection at high-toughness GBs.** We conducted our investigation on a 1 mm-diameter cylindrical sample of alloy 725 that had been electrochemically charged with H and loaded to failure in tension (Methods). The sample contains the tip of a large, intergranular secondary crack. Figure 1a shows the XRAT scan of our sample with the void volume within the secondary crack shown in black. All the fracture surfaces observed in the sample are interconnected: no independent cracks disconnected from the main crack were observed. We determine the average orientation of the fracture surface by manually fitting a plane to the crack within the core sample and find that this plane is not perpendicular to the sample axis. We attribute this deviation to the marked, non-uniform plastic distortion of the tensile specimen (Methods), which causes the tensile axis to drift away from the sample axis during the test. Similarly, we identify a nominal crack propagation direction as the direction within the nominal crack plane that lies perpendicular to the crack front. Figure 1b, c shows the crack along its nominal plane and parallel to its propagation direction. They show that the crack exhibits pronounced local deviations from its average orientation.

By investigating the morphology of the fracture surface in detail, we identified a set of crack deflection events (CDEs): local surface morphologies where the crack deviates markedly from the nominal fracture plane, even though a less tortuous path is available for the crack to propagate. Figure 2a shows an example of such a CDE. Here, and in the remainder of this work, the grains are stylized as 3D surface meshes of the segmented HEDM data and colored according to their average orientation relative to the laboratory reference frame. Part of the crack surface—labeled *n* for nominal in Fig. 2a—is aligned with a GB oriented parallel to the nominal fracture plane. Another part of the crack surface—labeled *d* for deflected—is aligned with a GB inclined at a high angle with respect to the nominal fracture plane. There is a GB to the left of the location where the crack surface orientation changes. The adjacent grains that form this GB are labeled G1 and G2.

Our interpretation of the events responsible for the formation of the crack surface configuration in Fig. 2a is as follows. The

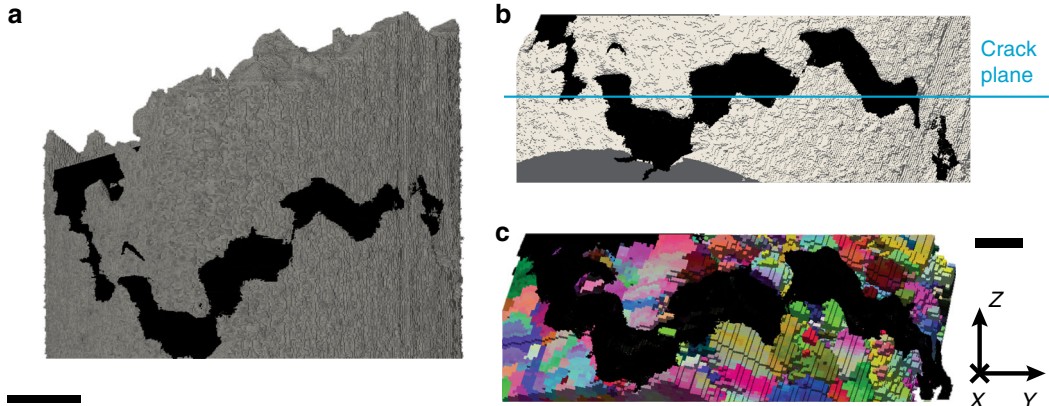

**Fig. 1** 3D sample reconstruction. **a** X-ray attenuation tomography (XRAT) data with metal shown in gray and void space inside the secondary crack shown in black. Scale bar: 200 μm. **b** Isolated section of the XRAT reconstruction with an edge-on view of the nominal crack plane (X–Y), indicated in blue. The nominal crack propagation direction (X) is into the page. **c** High-energy diffraction microscopy (HEDM) reconstruction of the microstructure viewed from the same direction as **b**. Each voxel is colored according to the crystallographic orientation of the crystal at that location relative to the laboratory reference frame. Scale bar for **b** and **c**: 100 μm. The black void space indicating the space inside the crack was obtained from the XRAT reconstruction and digitally fused with the HEDM data

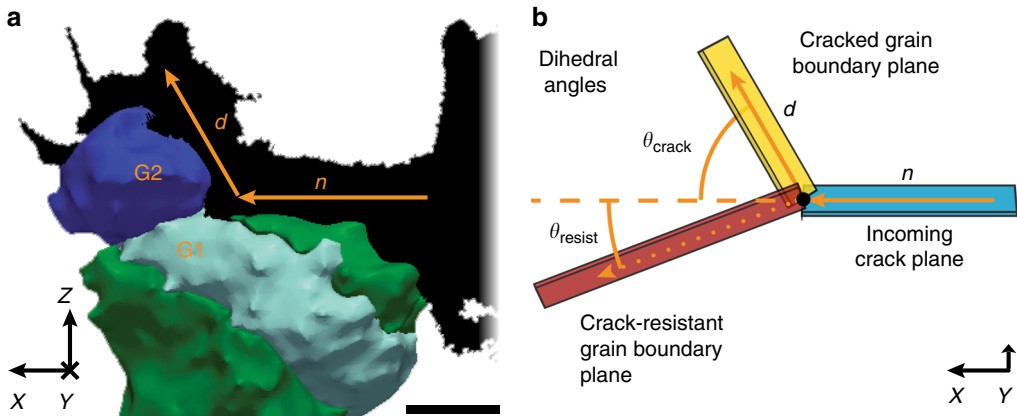

**Fig. 2** Crack deflection event (CDE). **a** CDE at the boundary between grains 1 and 2 (labeled G1 and G2, respectively). Scale bar: 50 μm. **b** Schematic illustrating the dihedral angles, $\theta_{crack}$ and $\theta_{resist}$, used to define our criterion for identifying CDEs. Traces of the incoming and deflected crack planes have been labeled in **a** with n and d, respectively

crack initially propagates from right to left along a single GB—generating crack surface n—until it reaches a triple-line between three grains. The GB between grains 1 and 2 is well aligned with the nominal crack plane orientation and therefore appears to be a favorable path along which the crack may continue to propagate. Nevertheless, the crack does not propagate along the boundary between grains 1 and 2. Instead, it proceeds along another GB inclined at a high angle with respect to the nominal crack plane, generating surface d. Thus, the crack propagates along the GB that—based on geometry—appears to be the less favorable of the two available intergranular fracture paths.

We developed a quantitative criterion for identifying CDEs based on the dihedral angle $\theta_{crack}$ between surfaces n and d and angle $\theta_{resist}$ between surface n and the plane of the un-cracked GB between grains 1 and 2, as illustrated in Fig. 2b. We define a CDE as a surface morphology where $\theta_{crack} > \theta_{resist}$. The incoming crack plane, n, and propagation direction are always assumed to be parallel to the previously described nominal crack plane and direction, respectively. Plane d and the un-cracked GB plane are fitted such that all three planes intersect at a common point along the crack front, which may be considered the decision point for the crack as it proceeds along one of the two available GBs. Both

of these planes are fitted manually to match as closely as possible with surface d and the un-cracked GB, neither of which is ever perfectly flat. In a comprehensive search through our HEDM/XRAT reconstruction, we identified ten CDEs. The dihedral angles, $\theta_{crack}$, and $\theta_{resist}$, are listed in Table 1. $\theta_{deflection}$, the difference between $\theta_{crack}$, and $\theta_{resist}$, indicates the magnitude of the deflection, and is also reported in the Table. The CDE illustrated in Fig. 2 is labeled as event #5 in Table 1.

It is surprising that CDEs should occur in situations where more favorably oriented crack propagation paths appear to be available. Indeed, if the crack propagation resistance of all GBs in the sample had been the same, then crack deflection would increase the work required to propagate a crack[53]. One possible reason for the occurrence of CDEs is that the local orientation of the tensile axis deviates markedly from the direction normal to the crack plane, n, favoring a change in crack propagation direction. However, the purpose of defining our CDE identification criterion with respect to an average nominal crack plane orientation is to maximize the likelihood that the tensile axis is indeed perpendicular to the crack plane. Thus, while we cannot exclude a priori that mixed mode loading affects the crack path, we believe that the likelier explanation for crack deflection is that

**Table 1 Dihedral angles $\theta_{crack}$, $\theta_{resist}$, and $\theta_{deflection}$ for the ten CDEs identified in our HEDM/XRAT reconstruction**

| Event | $\theta_{crack}$ (°) | $\theta_{resist}$ (°) | $\theta_{deflection}$ (°) |
|---|---|---|---|
| 1 | 57.1 | 35.7 | 21.4 |
| 2 | 53.2 | 35.4 | 17.8 |
| 3 | 61.2 | 35.4 | 25.8 |
| 4 | 115.5 | 79.2 | 36.3 |
| 5 | 64.6 | 20.7 | 43.9 |
| 6 | 72.9 | 18.8 | 54.1 |
| 7 | 63.2 | 39.8 | 23.4 |
| 8 | 47.8 | 21 | 26.8 |
| 9 | 59.2 | 49.7 | 9.5 |
| 10 | 68.7 | 39.1 | 29.6 |

**Table 2 Misorientation axes (defined in the crystal coordinate system of grain 1) and angles as well as the coincidence site lattice (CSL) designations for crack-resistant GBs in the ten CDEs found in our data**

| Event | Misorientation | | | | CSL Σ no. |
|---|---|---|---|---|---|
| | Angle, $\theta$ (°) | Axis | | | |
| 1 | 59.2 | 0.583 | −0.563 | 0.586 | 3 |
| 2 | 59.9 | 0.595 | −0.567 | 0.570 | 3 |
| 3 | 59.9 | 0.595 | −0.567 | 0.570 | 3 |
| 4 | 59.4 | 0.493 | 0.597 | −0.633 | 3 |
| 5 | 56.3 | 0.509 | −0.770 | −0.385 | − |
| 6 | 42.6 | 0.394 | −0.085 | 0.915 | − |
| 7 | 53.9 | 0.399 | −0.564 | −0.723 | − |
| 8 | 50.0 | −0.320 | −0.814 | −0.484 | − |
| 9 | 59.7 | 0.386 | 0.648 | −0.657 | − |
| 10 | 18.7 | −0.160 | 0.016 | −0.987 | − |

the boundary between grains 1 and 2 is more resistant to crack propagation than the GB aligned with surface $d$, along which the crack eventually propagates.

To substantiate this interpretation, we sought to determine whether the apparently crack-resistant GBs in our CDEs have any common characteristics. To this end, we used the HEDM reconstructions to determine their full crystallographic character. Table 2 reports the vector components of the misorientation axis (defined in the crystal coordinate system of grain 1) and the misorientation angle, $\theta$, for the high-toughness GBs ahead of the propagating crack in all ten CDEs. Based on this data, we assign Σ values to each GB according to the Brandon criterion[54,55]. Following the common distinction between special and general boundaries[14,56], only Σ values between 1 and 29 are reported. Four of the ten observed CDEs occur on Σ3 boundaries. Of the remaining six crack-resistant GBs, none were low-Σ boundaries ($1 < \Sigma \leq 29$) and none were low-angle GBs ($\theta < 15°$). Therefore, based on misorientation alone, all six would be classified as general GBs[14,56]. The character of the fractured GBs along the $d$ surfaces could not be identified due to the separation of the crack faces and the plastic deformation of the grains adjoining the crack surfaces, which results in low-confidence HEDM voxels in those grains.

Table 3 provides the plane orientations of the two grain facets that meet at the crack-resistant GB in each CDE. The plane normal directions are given in the Nye frames of each of the two crystals that meet at the GB. The lowest-index Miller plane that best matches the GB plane is also reported for each plane normal along with the deviation of this lowest-index plane from the true GB plane determined from the HEDM reconstruction. Out of the ten GBs reported in Table 3, nine have at least one grain with a

low Miller index facet along the GB plane. We define {001}, {011}, and {111} as low-index planes, and consider a GB plane to be low-index if it is within 10° of any member of these plane families.

For CDEs 1 through 4, both adjoining grain facets are {111} planes. Therefore, considering both Tables 2 and 3, we conclude that all four of the GBs in these CDEs are Σ3 CTBs. Consistent with previous studies on CTB traces along free surfaces[57,58], these boundaries show a few degrees deviation from ideal {111} facets. For the GB in CDE 5—which is illustrated in Fig. 2—the GB plane is within 9° of {001} in grain 1 and within 6° of {111} in grain 2. Both are low-index planes, according to our definition. The GB in CDE 6 has one low-index plane ({111} in grain 2) and one {123} plane (in grain 1), which does not qualify as low-index. Indeed, the only CDE with a crack-resistant GB that has no low-index facets is CDE 10. These findings suggest that the common feature of GBs resistant to H-assisted intergranular fracture is that they are most often boundaries with low-index planes: GBs with at least one grain facet along a low-index plane. Figure 3 summarizes the GB plane orientations in the ten CDEs we observed.

Following Seita et al.[21], we assess the statistical significance of our conclusion by computing its $P$ value. This analysis assumes the null hypothesis that there is no correlation between a GB's designation as a BLIP and its crack propagation resistance. It then computes the probability that nine out of ten CDEs nevertheless occur at BLIPs by dint of chance alone. To carry out this calculation, we first determine what fraction of all GBs in our sample are BLIPs. Using the data in ref. [59], which reports the GB character distribution in the sample investigated here, we find that 38% (by area) of all GBs in our sample are BLIPs. Using the binomial distribution, we find that the probability of nine or more out of ten CDEs occurring at BLIPs by chance alone—given that 38% of all GBs are BLIPs and assuming BLIPs are equally likely to deflect cracks as non-BLIPs—is 0.1% ($P = 0.001$). Consequently, we view the null hypothesis as having been falsified by our experiment and conclude that BLIPs are more resistant to crack propagation than non-BLIPs.

**Twin intersection-induced GB toughening**. Of the nine BLIPs listed in Table 3, four are CTBs. This finding is consistent with previous investigations[14], which concluded that twin boundaries —specifically CTBs[21]—reduce susceptibility to H-assisted propagation of pre-existing cracks in Ni and Ni-base alloys. Here, we find that, in addition to generating CTBs, twins may also reduce a material's susceptibility to H-assisted fracture in another way, namely by modifying the character of other GBs upon which they impinge.

We observed two instances (including CDE #6 in Tables 1–3) in which GBs were intersected by a twin lamella and the altered character of the GB at the twin intersection caused an intergranular crack to arrest or change course. Figure 4a shows a 3D reconstruction of one of these two cases (CDE #6). The crack initially propagates along the boundary between grains 3 and 4. However, at the boundary between grains 1 and 4 it becomes arrested and instead continues along a different, more tortuous path. To further illustrate this behavior, Fig. 4b provides a 2D section through Fig. 4a. This finding suggests that the boundary between grains 1 and 4 has elevated crack-propagation resistance compared to the boundary between grains 3 and 4, even though the alignment of both GBs with respect to the nominal crack plane is nearly identical.

The crack-resistant GB created by the twin lamella is not a low-Σ or low-angle boundary. However, by measuring its full five-parameter crystallographic character, we find that it is in fact a

**Table 3 Plane normal directions for the crack-resistant GB in each CDE, provided in the Nye frames of both adjoining grains. Lowest-index Miller planes that best match each GB plane are given along with the angle describing the deviation between the two**

| | Grain 1 | | | | Grain 2 | | | |
|---|---|---|---|---|---|---|---|---|
| Event | GB plane normal | | Lowest-index best match plane | Deviation (°) | GB plane normal | | Lowest-index best match plane | Deviation (°) |
| 1 | −0.509 0.634 −0.582 | | 111 | 5.1 | 0.653 0.563 −0.507 | | 111 | 6 |
| 2 | 0.548 −0.660 0.514 | | 111 | 6.2 | −0.611 −0.617 0.496 | | 111 | 5.5 |
| 3 | 0.548 −0.660 0.514 | | 111 | 6.2 | −0.611 −0.617 0.496 | | 111 | 5.5 |
| 4 | 0.631 0.485 −0.606 | | 111 | 6.4 | 0.607 0.621 −0.496 | | 111 | 5.6 |
| 5 | −0.115 −0.093 −0.989 | | 001 | 8.5 | 0.503 0.578 −0.642 | | 111 | 5.6 |
| 6 | −0.191 −0.574 −0.796 | | 123 | 5 | −0.621 −0.499 −0.604 | | 111 | 5.3 |
| 7 | 0.679 0.162 −0.716 | | 101 | 9.4 | 0.093 0.099 −0.991 | | 001 | 7.8 |
| 8 | −0.713 0.050 −0.700 | | 101 | 2.9 | 0.084 −0.214 −0.973 | | 001 | 13.3 |
| 9 | 0.722 −0.103 −0.684 | | 101 | 6.1 | −0.122 0.342 −0.932 | | 113 | 10.7 |
| 10 | −0.941 −0.262 −0.214 | | 311 | 5.9 | −0.811 −0.534 −0.240 | | 321 | 1.6 |

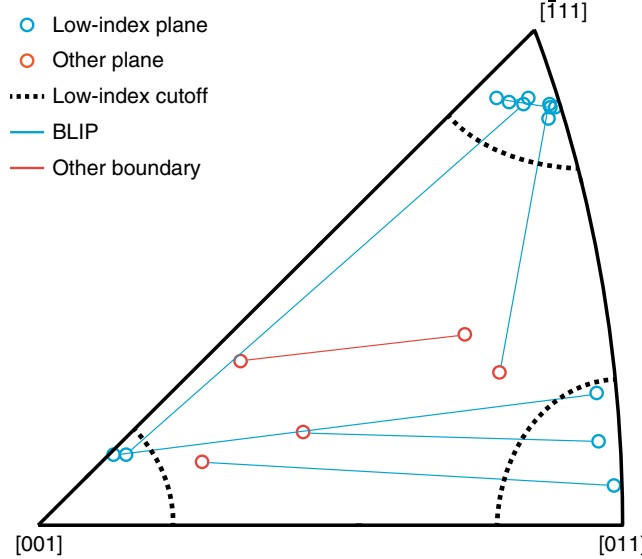

**Fig. 3** Crack resistant boundary facets. Orientation of plane normal directions for each pair of grains meeting at the un-cracked grain boundary in a crack deflection event (CDE). Plane orientations are expressed as Miller indices in one irreducible triangle of the stereographic projection. Lines connect corresponding planes at individual grain boundaries (GBs), with open circles representing plane normal orientations. Blue circles represent low-index planes, and red circles represent all other planes. Dotted lines indicate the 10° cutoff used to identify planes as low-index. Blue lines indicate boundaries with low-index planes (BLIPs), i.e., boundaries that contain at least one low-index plane, while the red line indicates the single CDE occurring at a GB without any low-index planes

BLIP. Thus, BLIPs may be generated by intersections of twins with other GBs. In the case of Fig. 4, the twin locally alters the GB character to transform part of the original boundary into a more crack-resistant one, ultimately toughening the microstructure. Figure 4c and d illustrates this effect schematically. This result demonstrates that a GB is as tough as its toughest part. Therefore, by increasing the variety of crystal facets on GBs, twin lamellae toughen these boundaries, on average.

**Frictional sliding and crack arrest in grain interiors**. Figure 5 demonstrates the occurrence of frictional sliding[53] in our sample: impingement of opposing crack surfaces upon each other followed by their sliding relative to each other along their area of

contact. This instance of frictional sliding occurred in CDE #4 from Tables 1 to 3. Here, the crack was deflected around grain 4 at a dihedral angle greater than 90°, leading to the impingement of grain 4 upon grain 5 during continued loading. The XRAT reconstruction in Fig. 5c shows contact between the two grains. The very-low-confidence index of the HEDM data in surrounding grains, shown in Fig. 5d, is consistent with extensive plastic deformation, as would be expected at surfaces that have undergone frictional sliding. Figure 5e and f shows a schematic illustration of the frictional sliding mechanism.

In addition to CDEs, we also observed two distinct cases (not shown) where the crack propagated into a grain interior and became arrested, and only a single case of transgranular fracture. These instances demonstrate that there are isolated regions within the microstructure where there are no low-toughness or favorably oriented GBs along which an intergranular crack may propagate. They also demonstrate that there is very little H-assisted transgranular fracture in these samples, as crack segments that are forced into grain interiors are likely to blunt and cease propagating. This observation reinforces the view that HE in alloy 725 is fundamentally connected to a change in fracture mode from transgranular to intergranular.

## Discussion

Our investigation of crack morphology in a H-embrittled sample of alloy 725 identified ten CDEs: instances where a high-toughness GB deflected the crack onto a tortuous, meandering path. By analyzing the crystallographic character of the ten high-toughness boundaries, we found that nine of them fit our definition of a BLIP, i.e., one where at least one of the grain facets that meet at the boundary have Miller indices within the {001}, {011}, or {111} families. We observed that twin lamellae impinging onto GBs may alter the crystallographic character of the intersected boundaries, converting them into high-toughness BLIPs. Finally, we found that cracks deflected more than 90° from the nominal crack path may lead to the re-impingement of the two opposing crack faces and subsequent frictional sliding between them.

These results illustrate several toughening mechanisms at work in H-charged alloy 725. The most apparent is crack meandering induced by CDEs. By forcing the crack path to become more tortuous, this mechanism increases the total surface area created by the advancing crack, elevating the total work required for fracture. Crack meandering may also improve toughness by locally reducing the crack driving force[53,60]. Another toughening mechanism caused by CDEs is frictional sliding, which opposes the opening of the crack. Frictional sliding is also thought to occur in fatigue loading, where it might oppose both the opening

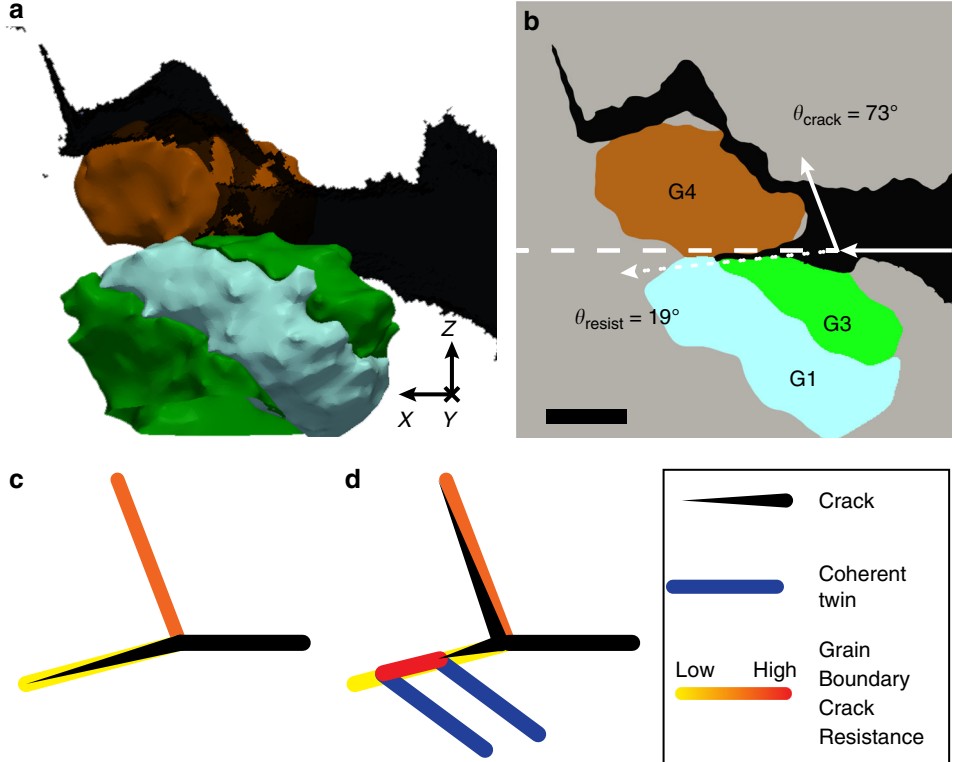

**Fig. 4** Twin intersection toughening. **a** Crack deflection event (CDE) #6 taking place at a boundary with low-index plane (BLIP) created by the impingement of a twin lamella on a grain boundary (GB). **b** A 2D section through the 3D reconstruction in **a** showing traces of the incoming, deflected, and grain boundary planes (see schematic Fig. 2b) as well as values of the dihedral angles, $\theta_{crack}$ and $\theta_{resist}$, used to identify the CDE. Solid arrows indicate the path of the crack, the dashed line indicates the nominal crack direction, and the dotted arrow indicates the path along the crack-resistant boundary that deflected the crack. The angles indicated are the dihedral angles defined in Fig. 2b. Scale bar: 50 μm. **c, d** Schematic showing the role of twin lamellae in altering the crack-propagation resistance of GBs. **c** A crack reaching a triple junction propagates along a low-toughness GB that is aligned well with the crack plane. **d** A twin lamella intersecting the GB locally alters the GB character, creating a BLIP that arrests the propagating crack and deflects it onto a different, more tortuous path

and the closing of the crack[53], and is comparable to contact shielding in ceramics[61]. The likelihood of frictional sliding increases with crack tortuosity, further emphasizing the importance of CDEs for the fracture behavior of alloy 725.

Both crack meandering and frictional sliding may be viewed as extrinsic toughening mechanisms, in that they are primarily due to changes in the geometry of the crack path[53]. By contrast, toughening by twin intersections with GBs may be considered intrinsic in that it increases a GB's inherent fracture resistance, regardless of crack path geometry. Evidence of toughening by twin impingement has been previously gathered in investigations of intergranular fracture of copper, where this mechanism is thought to cause major changes in crack-propagation resistance[62]. However, previous studies did not identify the full character of the twin-intersected boundaries and therefore did not connect toughening to the formation of BLIPs in the region of twin impingement. Our work also provides the first evidence of this mechanism in a H-embrittled sample.

A twin lamella impinging on a section of GB changes that section's misorientation while leaving the GB plane orientation in the laboratory frame unaltered. This GB section therefore has modified crystallographic character and may have higher or lower fracture toughness than before its intersection with the twin, depending on whether the twin converted a non-BLIP section into a BLIP or vice versa. However, as discussed in connection with Fig. 4, our work suggests that it is a GB's strongest section that determines its ability to arrest or deflect a crack. Thus, increasing the diversity of GB crystallographic characters along

any given GB plane is likely to elevate the crack propagation resistance along that plane, on average, militating in favor of increasing the density of twins in polycrystalline materials to improve fracture toughness.

All of the fracture surfaces in our alloy-725 sample are interconnected into a single, continuous crack. This finding suggests that there is no crack initiation in the interior of the sample, consistent with the view that all cracks initiate at free surfaces in Ni-base alloy 725[63] as well as in other materials[64]. It also indicates that there is no toughening by microcracking in H-charged alloy 725. In previous investigations[21,65], we ruled out that precipitates, such as carbides, play a role in H-assisted crack initiation or propagation in this material, demonstrating that GBs are the preferred sites for both crack initiation and propagation.

Numerous previous investigations ascribed high toughness to GBs with low Σ, as designated by the coincident site lattice (CSL) model[46], or low misorientation angles. The CSL model appears to be effective in capturing some trends in GB properties, such as GB energy, creep strength, fatigue strength, and intergranular corrosion resistance[66], prompting efforts to increase the fraction of low-Σ boundaries to improve the performance of polycrystalline solids[67]. For example, Bechtle et al. showed that increasing the length fraction of low-Σ GBs from 46 to 75% doubles the tensile ductility of H-charged pure, polycrystalline nickel, and reduces the fraction of intergranular fracture by 60–100%[14]. One limitation of models based on Σ numbers is that they cannot capture the full GB character: Σ numbers depend only on GB misorientation. Meanwhile, the importance of accounting for the

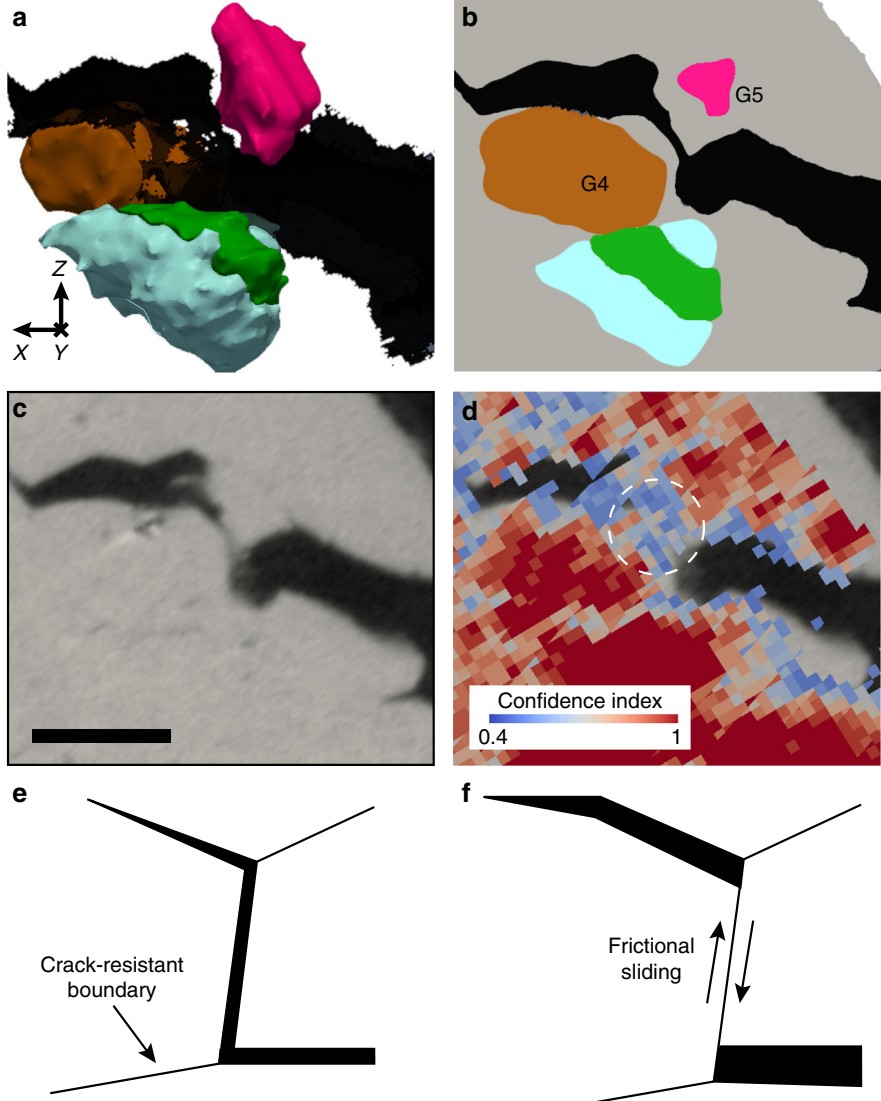

**Fig. 5** Frictional sliding. **a** 3D reconstruction of crack deflection event (CDE) #4 and **b** a 2D cross section showing the grains that undergo frictional sliding. **c** A 2D section through the corresponding x-ray absorption tomography data showing opposing crack faces in contact in the region around grains 4 and 5. Scale bar: 100 μm. **d** Low confidence index voxels in the high-energy diffraction microscopy reconstruction in the vicinity of the sliding contact (highlighted by the white circle), indicating severe local plastic deformation. **e, f** Schematic of frictional sliding mechanism. **e** A crack is deflected past vertical due to a crack-resistant boundary, leading to **f** the re-impingement of grains from opposing crack faces upon each other followed by sliding along their area of contact

full GB character, including the additional two degrees of freedom that describe the GB plane[52], is becoming increasingly clear. By identifying BLIPs as the GBs most resistant to H-assisted fracture, our work reinforces the need to account for GB plane orientations in GB property models and, consequently, further demonstrates the incompleteness of models based on Σ alone.

In a pioneering study, King et al. used diffraction contrast tomography to investigate intergranular fracture resistance as a function of the full GB character[68]. They identified three bridging ligaments along an intergranular stress corrosion crack in a sensitized austenitic stainless steel. Each of these ligaments contained one high-toughness GB. One of these GBs is a non-CSL boundary with {100} and {321} grain facets, qualifying it as a BLIP. Of the remaining two GBs, one is a low misorientation angle GB and the other a Σ11 boundary, but neither of them is a BLIP. These results are not sufficient to infer whether BLIPs, low-angle boundaries, or low-Σ boundaries are especially resistant to stress corrosion cracking. By comparison, the results presented here demonstrate

—with quantified statistical significance—that BLIPs are in fact more resistant to H-assisted fracture than non-BLIPs.

To determine what physical—rather than merely crystallographic—factors are responsible for the resistance of BLIPs to H-assisted fracture, it is instructive to consider the case of CTBs first. CTBs are Σ3 twin boundaries with GB planes aligned with {111} facets in both adjoining grains. CTBs are therefore BLIPs. Indeed, four of the nine BLIPs that were found to cause CDEs in our study were CTBs. This result is consistent with previous work, where we showed that CTBs are crack propagation resistant in H-charged alloy 725[21,69]. Indeed, CTBs are widely expected to be crack propagation resistant because they have lower formation energy (and therefore higher surface separation energy)[70] and lower H solubility[71] than other GBs. Recent density functional theory (DFT) calculations have furthermore demonstrated that CTBs in Ni have negative free volume[72].

BLIPs may be fracture resistant for the same reasons as CTBs. Randle has argued that BLIPs, like CBTs, are likely to have low

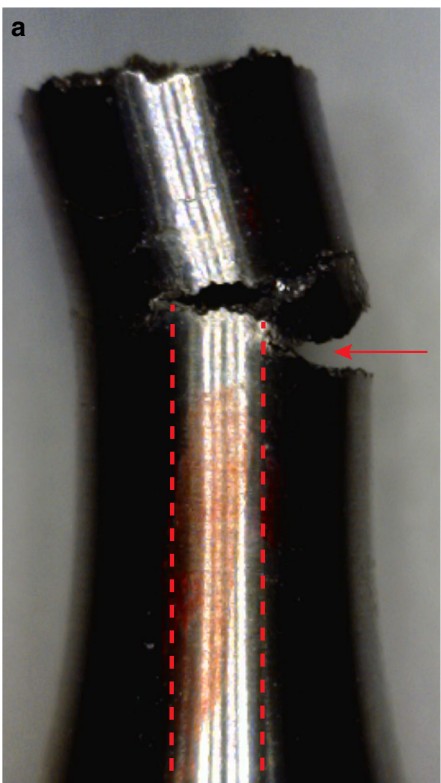
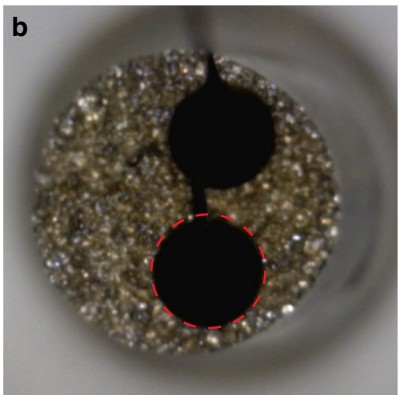
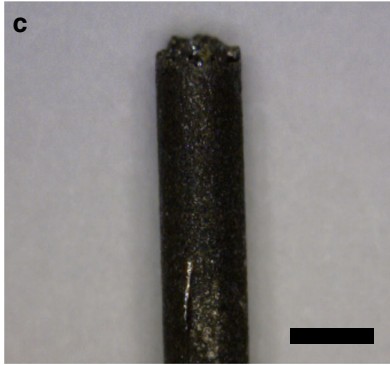

**Fig. 6** Sample preparation. Side (**a**) and top (**b**) views of the tensile specimen investigated in the present study. Red dotted lines indicate the approximate location of the core sample shown in **c**. The red arrow in **a** indicates the opening of a large secondary crack, on which the present investigation is focused. Scale bar in **c**: 1 mm

(albeit perhaps not negative) free volume[52]. Many may possess stepped surfaces with true low-index facets. Moreover, measurements of complete GB character distributions in polycrystalline Ni[73] as well as in alloy 725[59] find an increased frequency of occurrence of BLIPs, especially ones with {111} facets (even after CTBs have been excluded). The higher-than-random incidence of BLIPs in polycrystals demonstrates their comparatively low energy[47]. Finally, Zhou et al. showed that the solubility of H at GBs in Ni depends on the distribution of sites with distinct volumes and bonding geometries at the GB[74]. Additional work is needed to determine whether sites that favor low H solubility are especially scarce at BLIPs or whether other factors—such as interactions with dislocations—are responsible for BLIP fracture resistance.

While our data shows that BLIPs are resistant to H-assisted fracture, it does not allow us to conclude that non-BLIPs are preferential sites for crack propagation. Demonstrating the latter requires determining the character of GBs along which cracks propagated. However, as mentioned before, the grains found at crack surfaces are so heavily deformed that their orientations cannot be determined using HEDM. Therefore, to investigate the susceptibility of non-BLIPs to crack propagation, future studies should perform HEDM grain mapping both prior to and after mechanical testing. Then, the path ultimately taken by the crack may be correlated with the character of GBs in the initial, undeformed microstructure. Thanks to improved capabilities available at APS's 1-ID beamline[75], in situ studies examining the time-history of cracking under continuous loading are now possible[76,77]. A key additional development needed to enable in situ studies of H-assisted fracture is the capability to apply mechanical loads while simultaneously performing electro-chemical charging.

The identification of BLIPs as fracture resistant GBs has important potential technological implications. First, it opens the opportunity for improved lifetime predictions based on measurements of the number and distribution of BLIPs in individual components. Such predictions may be used to reduce the occurrence of unexpected failures by imposing service limits that are microstructure informed[69,78]. They may also be used to extend operating lifetimes by averting premature withdrawal of components from service due to overly conservative failure models. Second, our findings suggest new strategies for the design of HE-resistant materials via GB engineering[51], in particular by focusing effort on maximizing the number of BLIPs in the material, rather than of low-Σ GBs.

## Methods

**Sample preparation.** Tensile specimens 25 mm in length and with a 3.8-mm-diameter gage section were fabricated by the Centro Sviluppo Materiali laboratory from a forged bar of alloy 725 obtained from Special Metals. The reported ASTM grain size was 3–4, corresponding to average grain diameters of 90–125 μm. The specimens were electrochemically charged at 80 °C and −1.5 V for 7 days in 1 M deaerated NaCl prior to being tested to failure using slow strain rate testing (SSRT)[79]. They remained under these charging conditions during straining to maximize the effects of HE. The applied strain rate was $4 \times 10^{-6}\,\text{s}^{-1}$ and the final hydrogen concentration was reported to be 35 appm. Scanning electron microscopy characterization of the surface of the main crack indicates that the fracture of the tensile test specimen is almost entirely intergranular brittle-like. Significant secondary cracking was also observed, consistent with previous reports on HE in similar alloys[64].

In preparation for high-energy X-ray measurements, 1 mm-diameter cylindrical core samples were machined from the interior of one of the tensile test specimens using electrical discharge machining. The cylinder axis of these cores is parallel to the initial tensile axis of the test specimen, as shown in Fig. 6. These cores were taken to reduce the sample diameter to one amenable to scanning via the HEDM technique at beamline 1-ID of the Advanced Photon Source (APS) at Argonne National Laboratory. In addition, they served to isolate a region of the sample containing a secondary crack. The core sample studied here contains the crack tip

**Table 4 Mean and maximum deviations of voxel orientations from the average orientations of the grains that meet at the crack resistant GBs in each CDE**

| Event | Grain 1 | | Grain 2 | |
|---|---|---|---|---|
| | Mean (°) | Maximum (°) | Mean (°) | Maximum (°) |
| 1 | 0.35 | 3.7 | 0.98 | 3.9 |
| 2 | 0.03 | 2.1 | 0.56 | 5.2 |
| 3 | 0.03 | 2.1 | 0.56 | 5.2 |
| 4 | 0.16 | 5.9 | 0.81 | 10.0 |
| 5 | 0.36 | 3.8 | 0.19 | 4.4 |
| 6 | 0.36 | 3.8 | 0.54 | 6.1 |
| 7 | 0.27 | 5.7 | 0.05 | 2.7 |
| 8 | 0.11 | 2.7 | 1.04 | 4.6 |
| 9 | 0.59 | 5.6 | 3.30 | 9.1 |
| 10 | 0.43 | 7.6 | 1.00 | 6.1 |

of one branch of this secondary crack, which is the focus of our investigation. Because this crack does not travel through the entirety of the sample, it provides a view onto two mating fracture surfaces that have not yet fully separated, offering insights that cannot be obtained easily by investigating individual, fully exposed fracture surfaces.

**Acquisition and analysis of synchrotron data.** Samples were characterized using near-field HEDM (nf-HEDM)[80,81] and XRAT[76,82] at APS beamline 1-ID. Both techniques are non-destructive. The nf-HEDM technique generates a series of 2D maps of crystallographic orientations that are then integrated into a fully 3D reconstruction of the microstructure. Successive 2D maps are spaced 8 μm apart and each map is discretized on a grid of 8 μm equilateral triangles, resulting in a roughly isotropic 8 μm spatial resolution. The discretization of sample space, which is on a length scale much smaller than the grain size, results in measurement sensitivity to intragranular orientation variations. In total, 150 maps were acquired, such that the total height of the scanned volume was ~1.2 mm.

The reconstructed microstructure was visualized using the open source software package ParaView[83]. DREAM.3D[84] is employed to reconstruct the 3D GB network and characterize each GB by the lattice misorientation and boundary plane orientation. The orientation resolution of the nf-HEDM in this particular experiment was estimated as 0.1° in the vicinity of the GBs. GB misorientations are determined using the average orientations of adjacent grains. Table 4 reports the mean and maximum deviations of voxel orientations from the average orientations of grains found at the crack resistant GBs identified in our study. In all but one GB, the mean voxel deviation from the average grain orientation is ~1° or less, indicating that GB misorientations computed from grain average orientations are accurate to within about ~2°.

Once the 3D microstructure is reconstructed, the full GB character can be ascertained: three rotation angles to describe the misorientation between the two crystals that form the boundary and two spherical angles that specify the orientation of the boundary plane. The present analysis does not reconstruct the microstructure of the entire cylindrical sample, but rather a hexagonal prism circumscribed by it. Details on experimental procedures and data reconstruction methods for the nf-HEDM conducted here are reported in ref. 59, together with an analysis of the GB character distribution in our sample.

XRAT is a specialized computed tomography (CT) technique[85] conducted at the same beamline at APS as nf-HEDM, allowing the same volume to be scanned by both methods without disturbing the sample and enabling accurate alignment of the resulting data sets[86]. The resolution of the detector was 1.5 μm/pixel, which results in the same resolution in the CT reconstructions. This tomography data is used to detect sample surfaces, internal voids, and cracks, since nf-HEDM is not able to accurately reconstruct crack morphology due to lack of diffraction signal in the cracks. Therefore, the combination of the two data sets enables determinations of grain shapes and crystallographic orientations along fracture surfaces.

**Data availability**. The data sets generated and analysed during the current study are available at the Argonne National Laboratory's Materials Data Facility[87], https://doi.org/10.18126/M2063F.

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

## Acknowledgements

This work was supported by the BP-MIT Materials and Corrosion Center. This research used resources at the Advanced Photon Source, a U.S. Department of Energy (DOE) Office of Science User Facility operated for the DOE Office of Science by Argonne National Laboratory under Contract No. DE-AC02-06CH11357. Access to shared experimental facilities at MIT was provided by the MIT Center for Materials Science Engineering, supported in part by the MRSEC Program of the National Science Foundation under award number DMR-0213282. Work at CMU was supported by Department of Energy/Basic Energy Sciences grant DESC0002001. Computational support for this research was provided by grant TG-DMR130061 from the National Science Foundation's Extreme Science and Engineering Discovery Environment (XSEDE) advanced support program. A.B. acknowledges support from the National Science Foundation, Grant No 1150862, for support in HEDM data analysis. J.P.H. acknowledges the Department of Energy Office of Science Graduate Fellowship Program (DOE SCGF), made possible in part by the American Recovery and Reinvestment Act of 2009,

administered by ORISE-ORAU under contract no. DE-AC05-06OR23100. We are grateful to B. Blaiszik for assistance with data archiving.

## Author contributions

J.P.H. identified and analyzed CDEs from registered HEDM and XRAT reconstructions. Both J.P.H. and A.B. contributed to the registration of HEDM and XRAT data. A.B. performed microstructure reconstruction from raw HEDM data and analyzed the GB character distribution in the sample. J.L. assisted A.B. in the microstructure reconstruction, under the supervision of R.M.S. P.K. performed the X-ray experiments at APS, reconstructed the XRAT data, and guided and discussed the evaluation procedures and data registration with J.P.H. and A.B. Samples were prepared by J.P.H. under the supervision of S.G. M.J.D. conceived the study and supervised its execution. The manuscript was written by J.P.H. and M.J.D. All authors reviewed and approved the manuscript.

## Additional information

**Competing interests:** The authors declare no competing interests.

