## [Peer review File · Nature Communications]

Reviewers' comments:

Reviewer #1 (Remarks to the Author):

The major claim of the paper is that grain boundaries that are close to a low hkl index plane are more resistant to crack propagation in a hydrogen embrittled Ni-alloy. A quite novel experiment has been performed that characterises the planes on the fracture surface and in the vicinity of crack deflections. The claim is supported by the data. It is consistent with other studies of the effects of grain boundary plane on fracture resistance, such as stress corrosion cracking and thermal sensitisation. Suitable citations are provided for these earlier works. The observations for a hydrogen embrittled Ni-alloy are novel. They provide new data to support the generally expected relationship between grain boundary plane and fracture resistance, so there is less novelty in this point. Nonetheless, the findings will be of interest to the general community, and this paper is likely to be cited as it presents a fairly succinct and clear demonstration of this phenomenon.

The paper is quite clearly written, and the experimental methodology is sound and well described.

Reviewer #2 (Remarks to the Author):

This is a potential interesting paper that examines the effect of grain boundary character on crack deflection and hydrogen embrittlement. While the technique, 3D high energy diffraction, is interesting and coupled with x-ray imaging, the crack can be seen, I feel that the observations are still too qualitative to make a definitive conclusion on the effect of GBC, and thus, publication in nature communications.

the authors should get more statistics and work on getting better and more convincing 3D renderings of the grain orientations. The analysis and presentation could be better.

One important point is that the experiment is ex situ which makes making conclusions about crack growth and deflection tenuous at best. The authors should consider pausing the experiment and doing several scans to really show the evolution of crack growth. In situ experiments with 3D XRD and crack growth have already been done and should be doable.

Reviewer #4 (Remarks to the Author):

In the manuscript by Prof Demkowicz and colleagues, the relationship between H-assisted fracture and crystallographic orientation is studied in depth using the state-of-the-art techniques of synchrotron based high-energy X-ray diffraction microscopy (HEDM) and X-ray absorption tomography (XRAT). The most important conclusion is the existence of the high fracture resistant BLIPs, where at least one of the neighboring grains has a low Miller index facet along the grain boundary plane. The results are interesting and well presented, and the findings provide new ideas of the design of HE-resistant materials via grain boundary engineering. However, the following questions should be answered before acceptance for publication.

1. My first major concern is about the local crystal orientation measurement. I believe that the morphology of the fracture surface can be accurately measured using XRAT, but how about the crystal orientation? The orientation resolution of the HEDM experimental setup employed in the study should be introduced. Moreover, as mentioned in the manuscript, a significant crystal orientation gradient is observed close to the fracture surface, due to the severe plastic deformation. Thus it is wondered how the crystal orientation of the grain boundary is measured/calculated. Is it calculated from the average orientation of the whole grain, or really the local orientation? How precise is it? It would be nice if the authors can give a reasonable estimation. This will also provide a clue to tell if these deflection/misorientation/deviation angles

make sense in Tables 1, 2, and 3.

2. In this manuscript, BLIPs are defined to be boundaries where at least one of the neighboring grains has a low Miller index facet along the grain boundary plane, and Miller index facets are considered to be $\{001\}$, $\{011\}$, or $\{111\}$ families. Can it be argued that maybe $\{113\}$ or/and $\{123\}$ also belong to BLIPs?

3. This is a relatively minor point: In Figures 2a and 4b, although it has been stated that the dihedral angles are measured, from what are marked in the figures, it is rather misunderstanding that the angles are formed by the plane traces. It would be nicer if the angles can be better represented in these figures.

4. Another minor point: In the first paragraph in the Discussion session, it would be nicer to emphasize that H-assisted fracture is discussed here. In the current version, it reads very much like a general discussion over any fracture phenomenon, which may not be real.

Kai Chen

Reviewers' original comments shown in normal black lettering

Our responses shown in blue italics

Reviewer #1 (Remarks to the Author):

The major claim of the paper is that grain boundaries that are close to a low hkl index plane are more resistant to crack propagation in a hydrogen embrittled Ni-alloy. A quite novel experiment has been performed that characterises the planes on the fracture surface and in the vicinity of crack deflections. The claim is supported by the data. It is consistent with other studies of the effects of grain boundary plane on fracture resistance, such as stress corrosion cracking and thermal sensitisation. Suitable citations are provided for these earlier works. The observations for a hydrogen embrittled Ni-alloy are novel. They provide new data to support the generally expected relationship between grain boundary plane and fracture resistance, so there is less novelty in this point. Nonetheless, the findings will be of interest to the general community, and this paper is likely to be cited as it presents a fairly succinct and clear demonstration of this phenomenon.

The paper is quite clearly written, and the experimental methodology is sound and well described.

We are glad that the reviewer views our work as being suitable for publication in Nature Communications without further modification.

Reviewer #2 (Remarks to the Author):

This is a potential interesting paper that examines the effect of grain boundary character on crack deflection and hydrogen embrittlement. While the technique, 3D high energy diffraction, is interesting and coupled with x-ray imaging, the crack can be seen, I feel that the observations are still too qualitative to make a definitive conclusion on the effect of GBC, and thus, publication in nature communications.

We are glad that the reviewer finds merit in our work and hope that the responses given below address the reviewer's concerns.

the authors should get more statistics and work on getting better and more convincing 3D renderings of the grain orientations. The analysis and presentation could be better.

We sympathize with the referee's insistence on high quality statistics and rendering of data. Indeed, we have aspired to the highest standards in both categories. Unfortunately, the time and resources needed to carry out a HEDM study are so high, that collecting a very large number of data point is infeasible. Still, the number of data points that we have obtained (i.e., 10!) is larger than any comparable study done before (see discussion for details). In fact, some HEDM investigations on rare events (e.g., fatigue crack initiation) only obtain one data point!

Unfortunately, referee #2 does not provide any specific suggestions for improvement. Nevertheless, we have carried out significant further analysis and made additional improvements in response to the constructive feedback of referee #4. We draw the reviewer's attention to our responses to referee #4 for a summary of the changes we have made.

One important point is that the experiment is ex situ which makes making conclusions about crack growth and deflection tenuous at best. The authors should consider pausing the experiment and doing several scans to really show the evolution of crack growth. In situ experiments with 3D XRD and crack growth have already been done and should be doable.

The reviewer observes that in situ experiments may provide a more direct view of crack growth and deflection and that therefore the results presented in the current manuscript are "tenuous at best." We agree with the reviewer that in situ experiments have the potential to provide additional insights. Indeed, we already state the same conclusion in the discussion section of our original manuscript. However, we disagree that the unavailability of in situ experiments undermines our current results. On the contrary, the conclusions we have presented are supported by our data and we have carefully spelled out in detail exactly how we came to these conclusions.

Moreover, we believe that a thorough ex situ study such as the one we report here is a natural and necessary pre-requisite for future in situ studies, as it will inform choices of sample design, loading mode, data acquisition, etc. Indeed, a highly oversubscribed beamline like 1-ID at APS is unlikely to award beam time to an in situ study before an ex situ experiment such as the one we report has been carried out. Moreover, an in situ study will require a substantially longer beam time allocation than the one we used—an allocation that, in all likelihood, will not be awarded if the preceding ex situ study does not show significant impact.

The reviewer furthermore points out that in situ investigations of 3D crack growth have already been carried out and suggests that it is therefore already possible to perform in situ investigations of fracture under hydrogen embrittlement. Indeed, we are aware of previous studies that have tracked the evolution of crack morphology in 3D using HEDM. For example, reference 104 already cites one such study carried out at the 1-ID beamline at APS: the same beamline as the one we used in our work. In response to the reviewer's comments, we have now added three more references on in situ work at 1-ID.

However, it does not follow from the foregoing that in situ HEDM investigations of fracture under hydrogen embrittlement are also currently possible. For that, we would have to be able to both apply mechanical loads in situ and simultaneously perform electrochemical charging on the sample. This capability is not currently available at APS (or any of the synchrotron facility, to the best of our knowledge). In the revised manuscript, we have added a sentence in our discussion section to make it clear that the limiting factor for in situ studies of hydrogen embrittlement is the lack of a facility that would make it possible to perform simultaneous mechanical loading and electrochemical charging.

Parenthetically, we would like to inform the reviewer that Demkowicz and colleagues have recently submitted a white paper proposing to develop simultaneous mechanical loading and electrochemical charging capabilities at the 1-ID beamline at APS. Motivated in part by the reviewer's comments, we investigated what kinds of in situ stages are currently being developed at several facilities around the world and found that no stages currently under development are suitable for the in situ study that the reviewer suggested. This is hardly surprising, as making a safely operating hydrogen charging electrochemical chamber compatible with the near-field HEDM measurement setup is a substantial technical challenge. The sample diameter is necessarily small and the detector is only ~3-4 mm away from the center of the sample, which must be integrated into a load frame and surrounded by a corrosive medium. If our proposal is supported, then we will have the opportunity to tackle these challenges. However, such a development process will take significant time and effort, even if financial support is forthcoming. In all likelihood, no such in situ study will be performed by anyone anytime soon.

Reviewer #4 (Remarks to the Author):

In the manuscript by Prof Demkowicz and colleagues, the relationship between H-assisted fracture and crystallographic orientation is studied in depth using the state-of-the-art techniques of synchrotron based high-energy X-ray diffraction microscopy (HEDM) and X-ray absorption tomography (XRAT). The most important conclusion is the existence of the high fracture resistant BLIPs, where at least one of the neighboring grains has a low Miller index facet along the grain boundary plane. The results are interesting and well presented, and the findings provide new ideas of the design of HE-resistant materials via grain boundary engineering. However, the following questions should be answered before acceptance for publication.

We are glad that the reviewer considers our work acceptable for publication, provided that we address the constructive criticisms given below. We hope that our replies resolve the reviewer's concerns.

1. My first major concern is about the local crystal orientation measurement. I believe that the morphology of the fracture surface can be accurately measured using XRAT, but how about the crystal orientation? The orientation resolution of the HEDM experimental setup employed in the study should be introduced. Moreover, as mentioned in the manuscript, a significant crystal orientation gradient is observed close to the fracture surface, due to the severe plastic deformation. Thus it is wondered how the crystal orientation of the grain boundary is measured/calculated. Is it calculated from the average orientation of the whole grain, or really the local orientation? How precise is it? It would be nice if the authors can give a reasonable estimation. This will also provide a clue to tell if these deflection/misorientation/deviation angles make sense in Tables 1, 2, and 3.

We thank the reviewer for these constructive observations. In response, we have carried out substantial additional analysis of the orientations of the grain pairs that form the crack resistant GBs identified in our study. We have also expanded our discussion in the methods section ("Acquisition and analysis of synchrotron data" subsection) to convey the results of this analysis. Below is a point-by-point summary of this discussion:

- The orientation resolution of the nf-HEDM in this particular experiment was estimated as 0.1 degrees in the vicinity of the GBs.*
- GB misorientations are determined using average grain orientations.*
- We have analyzed orientation variations within each of the grains adjacent to the crack resistant GBs identified in our study. We did this by computing the deviation in voxel orientations from average grain orientations. The figure below shows a typical histogram of the angles by which individual voxels deviate from average grain orientations (data are shown for both grains forming the crack resistant GB in CDE 6). We see that the overwhelming majority of voxels have angular deviations of no more than 1 degree and no voxels have deviations of more than 6 degrees.*

We have constructed similar histograms for all the remaining CDEs, as well. To summarize our results, we have computed the average and maximum voxel orientation deviations from average grain orientations for all grains in all CDEs. A new table (Table IV) has been provided to report this data. In all but one GB, the mean voxel deviation from the average grain orientation is $\sim 1^\circ$ or less, indicating that GB misorientations computed from grain average orientations are accurate to within about $\sim 2^\circ$.

The reviewer also points out that grain orientations vary significantly in the vicinity of fracture surfaces, making it difficult to determine the misorientations of fractured boundaries. Indeed, as an example, in Fig. 5 (which depicts frictional sliding), the misorientation between G4 and G5 is challenging to determine with precision due to severe plastic deformation of the surfaces and as a result we do not report it. However, our conclusions throughout the manuscript rest on the intact grain boundaries (e.g., between G1 and G2 in Figure 2a), not on the fractured ones. We draw the reviewer's attention to our statements concerning this point in paragraph 6 of the Results section of the original manuscript.

2. In this manuscript, BLIPs are defined to be boundaries where at least one of the neighboring grains has a low Miller index facet along the grain boundary plane, and Miller index facets are considered to be $\{001\}$, $\{011\}$, or $\{111\}$ families. Can it be argued that maybe $\{113\}$ or/and $\{123\}$ also belong to BLIPs?

This is an excellent question and we thank the reviewer for bringing it up. Our definition of BLIPs is intended to account for the largest possible fraction of CDEs while using the smallest possible fraction of grain boundary character space. For example, had we broadened BLIPs to include $\{113\}$ and $\{123\}$ plane families, then our definition would have accounted for all 10 CDEs (rather than just 9). However, the fraction of all the grain boundaries in the microstructure that would

then qualify as BLIPs would have been 0.77: much higher than the 0.38 fraction under the original definition, resulting in substantial weakening of any conclusions that can be made concerning fracture resistance.

We believe that, ultimately, the findings in the present experimental study should serve as the basis for a more detailed follow-on investigation of the physical (rather than merely crystallographic) factors that give rise to HE resistance of BLIPs. The goal of such a study would be to explain the fracture behavior of BLIPs in terms of properties such as their energies, H solubility, structure, or interaction with dislocations. Indeed, we are currently conducting such a follow-on study using atomistic modeling methods. These points are already mostly addressed in two paragraphs in the discussion section (the 3rd- and 4th-to-last ones). However, in response to the reviewer's comments, we have made a few additions to these paragraphs to further emphasize the need for investigations of the physical mechanisms responsible for the fracture behavior of BLIPS.

3. This is a relatively minor point: In Figures 2a and 4b, although it has been stated that the dihedral angles are measured, from what are marked in the figures, it is rather misunderstanding that the angles are formed by the plane traces. It would be nicer if the angles can be better represented in these figures.

We sympathize with the reviewer's suggestion of presenting 3-D representations of GB and crack planes in Fig. 2.a) and 4.b), rather than their traces. While we have not found an adequate way to do this, we acknowledge that the original representation may appear confusing to some readers. Therefore, to mitigate this risk, we have made the following changes:

- Fig. 2.a): removed the GB trace and angle labels. Our hope is that this way we will avoid giving the impression that angles are measured between plane traces.
- Fig. 2.b): we have tried to represent the GB plane and dihedral angles in a more 3-dimensions way in this figure with the hope that readers will be able to correlate it with Fig. 2.a). We have also labeled the crack planes in both a) and b) to help in this correlation.
- We have expanded the caption of Fig. 2 to emphasize that the angles illustrated in b) are dihedral angles and explaining the relations between a) and b).
- Fig. 4.b): labeled the dihedral angles θ_{crack} and θ_{resist} , but without associating them with arcs that may mislead readers into associating these angles with plane traces. We have also expanded the caption to Fig. 4 to emphasize that the angles in b) are dihedral angles.

4. Another minor point: In the first paragraph in the Discussion session, it would be nicer to emphasize that H-assisted fracture is discussed here. In the current version, it reads very much like a general discussion over any fracture phenomenon, which may not be real.

We thank the reviewer for pointing out this ambiguity. To correct it, we have added a statement to the beginning of the paragraph in question clarifying that we are investigating a sample of alloy 725 subjected to hydrogen embrittlement.

REVIEWERS' COMMENTS:

Reviewer #2 (Remarks to the Author):

The authors have addressed most of the comments brought up by this reviewer. I still believe that the impact of this work is not as high as it could have been with in situ testing but it is a good step in that direction.

Reviewer #4 (Remarks to the Author):

The authors have answered all my questions, and I believe the manuscript is ready for publication.